# Effects of Nanomaterials on Crops

**DOI:** 10.3390/cimb47121024

**Published:** 2025-12-08

**Authors:** Xiaofang Yang, Huilian Xu, Wenrui Li, Xianchao Chang, Xiaohan Jiang, Xiaoyong Liu, Mengmeng Kong

**Affiliations:** School of Biological Science and Technology, University of Jinan, Jinan 250024, China; 202321100304@stu.ujn.edu.cn (X.Y.); 042172104438@ujn.edu.cn (H.X.); 19193284703@163.com (W.L.); 202321201484@stu.ujn.edu.cn (X.C.); 202421201572@stu.ujn.edu.cn (X.J.); bio_liuxy@ujn.edu.cn (X.L.)

**Keywords:** nanomaterials, crops, growth, ecological risk

## Abstract

It is essential to review the outcome of nanomaterials on crop growth and development, as well as their underlying applications and risks in agriculture. The agricultural department need to seek a more efficient and sustainable production approach with the global population growth and increasing resource pressure. A number of results have demonstrated that nanomaterials have significant advantages in the enhancement of crop stress resistance, promoting growth and yields due to their unique physical and chemical properties. The paper summarizes the impact of nanomaterials on seed germination, vegetative, and reproductive growth by analyzing existing research. It discusses how nanomaterials improve crop adaptability by regulating the antioxidant system, enhancing photosynthesis, and optimizing nutrient absorption. In addition, the review highlights the potential risks associated with nanomaterials in soil ecosystems, food chain transmission, and human health, including possible negative impacts on soil organisms, microbial communities, and food safety. Finally, this review emphasizes the need for the enhancement of long-term ecological security assessments and the development of intelligent delivery systems in future research, which can ensure the safe and efficient application of nano-agricultural technologies.

## 1. Introduction

The world population is expected to reach 9.6 billion, necessitating a 70–100% increase in agricultural production to fulfill escalating food requirements by 2050 [1,2]. Urbanization and industrialization have successively diminished arable land, while water scarcity has intensified [3]. Meanwhile, recurrent extreme climate events like droughts and cold waves have heightened abiotic stress on crops [4]. In agricultural production, the inefficient utilization of conventional chemical fertilizers and pesticides results in resource and disrupts the ecological balance of farmland [5]. This disruption exacerbates biological stress such as fungal diseases and insect pests, creating a disadvantage cycle that eventually reduces crop yields. Therefore, it is necessary to figure out novel approaches to enhance stress tolerance of crops and promote yield to resolve the requirements of global food supply.

Nanotechnology, across fields, can potentially protect plant growth and enhance plant tolerance to various stresses [6]. Nanomaterials (NPs) are materials ranging in size from 1 to 100 nm, possessing unique physical, chemical, and biological properties, such as high specific surface area, strong adsorption capacity, and adjustable surface charge. These capability make nanomaterials highly valued and widely applicable in agriculture. NPs, due to their minute size, can easily infiltrate plants and be transported to distant tissues from roots to shoots. Consequently, these NPs can accumulate at specific sites within plants and exhibit distinct biological functions [7]. Research indicates that silicon nanoparticles (Si-NPs) can improve drought tolerance through stimulating osmotic regulator synthesis and triggering antioxidant defense systems in barley [8]. Additionally, Hezaveh and Al-Amri et al. [9] discovered that the application of zinc oxide (ZnO) nanoparticles can enhance antioxidant enzyme activity under salt stress, facilitate osmotic regulator synthesis, and improve ion balance, effectively inducing salt-induced damage in rapeseed. These studies elucidate the potential mechanisms by which NPs enhance crop resilience to environmental stress through molecular regulation.

Through its broad prospects, nano-agricultural technologies still face key scientific challenges. Firstly, NPs of different types and concentrations induce various effects on plants, and the concentration limits of their beneficial and harmful effects have not been clearly investigated. Secondly, NPs’ migration, transformation, fate in the environment, and long-term ecological effects still need to be systematically assessed. They have attracted attention for their potential effects on plant growth, development, and even the entire food chain, with the accumulation of NPs in the environment [10]. The resolution of kernel challenges in nano-agricultural technology is closely linked to environmental safety and the industrial process. This review have systematically summarizesd the effects of various nanomaterials on key stages of crop development, including seed germination and asexual and sexual growth. It discusses the physiological and biochemical mechanisms that are affected by NPs. Additionally, this paper evaluates the potential ecological and health risks associated with the NPs in agriculture, which aim to provide a theoretical foundation and new ideas for their safe and efficient application.

## 2. Relevant Sections

### 2.1. Effects of NPs on the Growth and Development of Crops

Due to the differences in characteristics of nutrient absorption across different plants and different growth stages, the effects of various NPs on crops also show obvious stage specificity. Table 1 summarizes the effects of various NPs on the key stages of crop seed germination, vegetative growth, and reproductive growth.

#### 2.1.1. The Effect on the Stage of Seed Germination

Seed germination is the initial and critical stage in the growth of crops. In recent years, there has been growing interest in the role of NPs in this process [11]. Nanoparticles can penetrate the seed coat, effectively enhancing the seeds’ ability to absorb water and nutrients due to the small particle size. This mechanism of direct regulatory allows seeds to quickly obtain the necessary water and nutrients during the early stages of germination, which promote the germination process [25]. Research has shown that silver nanoparticles (AgNPs) promote seed germination through three mechanisms: (i) AgNPs create micropores on the seed coat surface, enhancing water uptake [26]. (ii) AgNPs are internalized into the seeds and upregulate the expression of aquaporin genes, improving transmembrane water transport. Aquaporins also facilitate the diffusion of reactive oxygen species (ROS) such as hydrogen peroxide (H_2_O_2_). Within a controlled “oxidative signaling window,” these ROS act as key signaling molecules involved in processes like cell wall loosening and endosperm softening, thereby promoting germination. (iii) NPs that enter the endosperm can function as nano-catalytic carriers. They bind to α-amylase via thiol bonds, forming complexes that induce favorable conformational changes in the enzyme. This enhances α-amylase activity, significantly accelerating starch hydrolysis and increasing the production of soluble sugars to support embryo growth [27].

Seed germination is the critical starting point for the growth of higher plants, which is particularly sensitive to environmental stresses. Adverse conditions can lead to a lower germination rate and seedling activity. Research indicates that treatments of various NPs can significantly enhance the germination performance and physiological activity of crop seeds under different stress. For example, treating lentil seeds with 10 μg/mL of Ag NPs can notably increase their germination rate, reduce the average germination time, and improve plant tolerance to drought stress [28]. Almutairi’s work also shows that treatments with Ag NPs lead to a significantly higher germination rate, seedling length, fresh weight, and dry weight of tomato seeds than those of the control group when exposed to salt stress [29]. This enhancement likely improves the crop’s resistance to abiotic stress due to NPs strengthening the physical barrier of the seed coat and activating the antioxidant defense mechanism [30].

Meanwhile, studies have indicated that ZnO NPs can function as an efficient seed-soaking agent for rice, significantly enhancing seed germination and starch metabolism. Treatment with 20 mg/L ZnO NPs resulted in over 50% increases in dry weight, relative germination rate, and root length of rice seedlings, along with a 23% rise in total soluble sugar content, a 45% boost in α-amylase activity, and a notable improvement in antioxidant enzyme activity. The mechanism of action of ZnO NPs is consistent with the aforementioned mechanism [31]. Furthermore, as a component of various key enzymes, ZnO NPs are involved in the synthesis of substances such as chlorophyll and proline. Under salt stress conditions, treatment with ZnO NPs increases chlorophyll and phenolic compound content in tomato seeds, reduces lipid peroxidation, maintains cellular osmotic balance, and raises the germination rate by 36.5% [32].

Carbon nanotubes enhance the water absorption capacity of tomato seeds through aquaporins and increase the concentration of elements such as Ca and Fe via cation exchange in the cell wall matrix, thereby promoting germination and plant growth [33]. In the range of 10–100 mg·L^−1^, fullerenes improve the seed germination rate, shoot dry weight, and photosynthetic efficiency of rapeseed under drought stress by regulating the ABA signaling pathway and enhancing ROS scavenging capacity [34].

However, high concentrations of NPs may inhibit seed germination and seedling growth. For instance, high concentrations of Ag NPs can hinder enzyme–substrate interactions or disrupt the catalytic structure of enzymes, reducing α-amylase activity [35] and thereby suppressing seed germination. ZnO NPs at 1000 mg/L decreased the germination rate of hibiscus seeds by 31.71% [36], as excessive nanoparticles may induce lipid peroxidation, hormonal imbalance, and seed dormancy [37]. Similarly, 1000 mg/L of graphene and carbon nanotubes reduced tomato seed germination rates by 19%, respectively [38], while 2000 mg/L of graphene further inhibited seedling biomass as well as root and shoot elongation [39]. The primary mechanism involves excessive NPs inducing ROS accumulation, leading to oxidative stress that subsequently suppresses germination and growth [38].

In summary, the suitable concentration of NPs can promote the germination of plant seeds and enhance the resistance to the biotic and abiotic stresses. However, if the concentration is out of the appropriate on plants, it may also inhibit the germination of seeds.

#### 2.1.2. The Effect on the Vegetative Stage of Growth

The vegetative growth stage of plants is the period when they primarily grow and develop roots, stems, leaves, and other vegetative organs after seed germination.

As the primary organ responsible for water and nutrient absorption, root progress plays a crucial role in crop growth and stress resistance [40]. Recent studies have demonstrated that nanomaterials have significant potential for enhancing root structure and function. The promoting effects of NPs on root architecture (including increased root length, density, and surface area) are not a singular outcome but are mediated by the complex interplay of multiple physiological and molecular mechanisms. Key mechanisms include regulation of phytohormone homeostasis, induction of controlled oxidative stress and signaling, and enhancement of nutrient availability and uptake.

Regulation of Phytohormone Homeostasis: Plant hormones are master regulators of root development [41]. NPs can directly or indirectly influence their synthesis and distribution. For instance, studies have shown that treatment with cerium oxide nanoparticles (CeO_2_ NPs) increases the levels of auxin and cytokinin in carrot plants by 22.2% and 33.7%, respectively. This hormonal shift stimulates the division of cambium cells in the central stele, leading to thickening of the primary root [42]. Similarly, titanium dioxide nanoparticles (TiO_2_ NPs) have been reported to promote lateral root formation in *Arabidopsis thaliana* and cucumber by interfering with auxin signaling pathways, thereby mimicking the effect of auxin itself [43,44].

Induction of Controlled Oxidative Stress and Signaling: While excessive reactive oxygen species (ROS) are detrimental, controlled bursts can act as signaling molecules that facilitate growth. Nanoscale zero-valent iron (NZVI) promotes root extension by inducing the generation of hydroxyl radicals (·OH). These radicals mediate the non-enzymatic scission of polysaccharides in the cell wall, leading to cell wall loosening and relaxation, a prerequisite for cell expansion [45]. This process enhances root metabolic activity and nutrient uptake.

Enhancement of Nutrient Availability and Uptake: NPs can improve root access to essential nutrients. First, certain nanoparticles, such as ZnO NPs and Fe_3_O_4_ NPs, can themselves serve as slow-release micronutrient sources [13]. Second, they can enhance the solubility and bioavailability of soil nutrients. For example, TiO_2_ NPs have been found to improve plant uptake of nitrate, potassium, and ammonium [46]. Furthermore, by promoting root elongation and increasing the total surface area and volume of the root system—as observed in rice treated with specific NPs [47]—NPs significantly improve the efficiency of water and nutrient acquisition from the soil.

However, the impact of NPs is highly dependent on the concentration. For instance, treatments with AgNPs have shown concentration-dependent effects: A concentration of 30 μg/mL promoted rice root branching and increased dry weight, whereas 60 μg/mL inhibited root growth [48]. Similarly, Li et al. found that 20 mg/L γ-Fe_2_O_3_ NPs significantly enhanced corn root length growth by 11.5%. In contrast, higher concentrations of 50 mg/L and 100 mg/L γ-Fe_2_O_3_ NPs reduced root length by 13.5% and 12.5%, respectively [49].

These interconnected mechanisms collectively explain how NPs at appropriate concentrations, improve crop root growth and physiological functions. Therefore, they ultimately lead to optimized root architecture and regulated hormone balance. Consequently, this helps mitigate environmental stress on plants.

The plant stems and leaves on the ground provide the mass and energy necessary for integral growth and development, mainly through photosynthesis and respiration during the middle growth stage. NPs can induce various effects, particularly on photosynthetic performance and stress resistance by regulating these processes. Photosynthesis is essential for stem and leaf growth with carbohydrate accumulation. Research has shown that the application of copper oxide nanoparticles (CuO NPs) in the soil enhances antioxidant enzyme activity of sugarcane and increases chlorophyll content [50]. Additionally, iron-based NPs, such as iron oxide nanoparticles (FeO NPs), uniquely influence the regulation of stomatal opening and closing. Specifically, treatment with 0.1 g/L of FeO NPs into the soil can promote the expression of the key gene AHA_2_ by five times, which regulates stomatal opening in *Arabidopsis thaliana*. The results suggest increased stomatal opening, enhanced CO_2_ absorption, and a greater photosynthetic leaf area [45]. Furthermore, treatment with 500 mg/kg of FeO NPs significantly improved assimilation rates and intercellular concentrations of CO_2_, as well as transpiration rates and stomatal conductance in plants, which can enhance photosynthetic efficiency [51]. The mechanism for regulating stomatal function also promotes the accumulation of carbohydrates such as glucose, sucrose, and starch, ultimately increasing the total biomass of the plants [52]. Importantly, the ability of NPs to enhance such fundamental physiological processes also equips plants with greater resilience to cope with various abiotic stresses.

NPs play a positive role in reducing abiotic stress in plants. For instance, treatment with ZnO NPs can relieve the negative effects of salt stress on wheat growth and its photosynthetic pigments [53]. Additionally, ZnO NPs enhance plant tolerance to drought stress by improving nitrogen metabolism, regulating stress-related proteins, increasing the activity of antioxidant enzymes, and stabilizing photosynthetic pigments [13].

It is important to call attention, however, to the fact that the effects of nanomaterials on plant stems and leaves are also highly dependent on their concentration. Leaf application of high concentrations of cerium oxide (CeO_2_) nanoparticles may prevent stomata, which can reduce transpiration and photosynthesis, ultimately inhibiting growth and resistance [54]. In contrast, low concentrations of CeO_2_ nanoparticles positively influence stomatal conductivity and photosynthesis [55]. Low concentrations (10–50 mg/L) of single-walled carbon nanotubes (SWCNTs) can stimulate cell growth in *Arabidopsis* mesophyll cells. However, at higher concentrations (100 mg/L), SWCNTs can lead to the accumulation of reactive oxygen species (ROS), which can cause cell necrosis and apoptosis [56].

In summary, nanomaterials can effectively enhance plant photosynthesis and alleviate abiotic stress at appropriate concentrations, providing the necessary material and energy support for stem and leaf growth. However, the effect is concentration-dependent, so reasonable application is the key to achieving its positive impact.

#### 2.1.3. Effect on Plant Reproductive Growth Stage

During the sexual reproduction of flowering plants, pollen grains play a crucial role in delivering male gametes [57]. After germination, they form of pollen tubes, which are the polar extension of the male gametophyte. These tubes pass through the stigma and style tissues, which grow directionally towards the embryo sac and allow for the precise transport of sperm cells to the female gametophytes [58]. Pollen germination is essential for initiating this transport process, and the successful elongation of pollen tubes is necessary for double fertilization to occur. In this process, two sperm cells must fuse with the egg cell and polar nuclei, respectively, to complete fertilization. Thus, pollen germination and pollen tube elongation together provide the structural and functional foundation for sperm transportation in the sexual reproduction of flowering plants, directly facilitating the success of fertilization.

To systematically summarize the effects of NPs on plant reproductive growth, this subsection begins with a comprehensive overview. Table 2 compiles the impacts of various NPs on pollen viability, pollen tube growth, and overall reproductive development.

As shown in Table 2, NPs exhibit a significant bidirectional regulatory effect on plant reproductive growth. Currently, research on the effects of NPs on pollen is limited, but existing studies have shown both inhibitory and beneficial effects. On one hand, certain NPs exhibit significant inhibitory effects. For instance, ZnO NPs inhibit the germination and tube elongation of lily pollen, while not affecting pollen viability [59]. Similarly, graphene oxide (GO) can significantly hinder pollen germination and elongation due to its acidic properties, resulting in abnormal pollen tube morphology, which may negatively impact plant sexual reproduction [60]. On the other hand, some NPs demonstrate positive effects. For example, Walid’s study found that foliar spraying of silver nanoparticles (Ag NPs) can significantly enhance the reproductive development of peach trees [61]. This enhancement is evident in the increased number of pollen grains, improved pollen viability, and the absence of malformation. Additionally, there is an increase in the number of spores and a broader distribution range, which helps improve the efficiency of pollination and fertilization, offering a new avenue for crop genetic improvement. Therefore, it is essential to consider the impact of NPs on pollen during the flowering phase.

The impact of NPs on the reproductive growth of crops is closely linked to both the yield and quality of produce, which result in increased interest in their use as fertilizers in agriculture. For example, zinc (Zn) is an essential nutrient for plants. It promotes photosynthesis, carbohydrate, and phosphorus metabolism and ultimately enhances grain development [63]. However, the validity of zinc can be significantly reduced by the presence of iron and aluminum oxides, clay minerals, and organic matter in the soil, which tend to adsorb and immobilize the zinc ions. In addition, excessive zinc fertilizer that is not absorbed by crops can damage the ecosystem. In contrast to traditional zinc fertilizers, nano-zinc fertilizers are less affected by soil texture, structure, and colloidal components, which makes them more easily absorbed and utilized by plants [64]. Studies have shown that applying of ZnO NPs can significantly increase rice yield compared to control treatments. Specifically, the number of spikelets per panicle increased by 4.81–10.69%, the 1000-grain weight grew by 3.82–6.62%, and the seed setting rate rose by 0.28–2.36%. Furthermore, at the dry jointing, heading, and maturity stages, the dry matter accumulation of rice treated with ZnO NPs was significantly higher than that of untreated rice. ZnO NPs also improved rice quality, enhancing the brown rice rate, milled rice rate, head milled rice rate, chalky grain rate, and protein content. This effectively optimized both the processing and appearance quality of the rice [64].

In addition to nano-zinc fertilizer, other NPs demonstrate excellent effects on promoting crop growth. A study by Kole et al. [62] found that seeds with fullerenol treatments, which has a particle size of 0.943–47.2 nm, could increase the biomass of bitter melon by up to 54%. The promotional effects spanned multiple aspects of growth, yield, and quality. Specifically, the fruit size increased by 20%, the number of fruits increased by 59%, the weight of individual fruits rose by 70%, and the total yield surged by 128%. In terms of active ingredients, inulin content increased by 91%, momordicin increased by 20%, lycopene increased by 82%, and cucurbitacin-B increased by 74%. All findings indicate that fullerenol not only promotes the growth and yield of bitter gourd, but also enhances its nutritional and medicinal value. Additionally, the foliar application of nano-calcium significantly increased the content of monomer anthocyanins, total phenols, and antioxidant activity in strawberries [65], further proving the potential of NPs in regulating crop quality.

In summary, NPs have a significant two-way regulatory effect on the sexual reproduction of flowering plants. On one hand, they may inhibit pollen germination and pollen tube elongation, disrupting the fertilization process. On the other hand, NPs, as a new type of fertilizer, show great potential in promoting crop growth, increasing yield, and improving quality. This includes enhancing pollen viability, increasing dry matter accumulation, optimizing fruit traits, and enriching bioactive components.

### 2.2. Nanomaterials to Plants: Transport, Oxidative Stress, and Photosynthesis

As the application of nanotechnology in agriculture continues to grow, the effects of nanomaterials on plants have gained significant attention. While these materials may enhance crop yield and quality, it is essential to understand their behavior in plants. This includes determining whether NPs can enter plant tissues, how they move within the plant, and where they accumulate. NPs can disrupt the oxidative balance and lead to oxidative stress, when NPs reach a certain concentration in plants. In response, plants activate their antioxidant systems, such as superoxide dismutase and glutathione, to protect themselves [66]. Furthermore, photosynthesis, which is crucial for plant growth, is sensitive to environmental changes. NPs can influence the synthesis of photosynthetic pigments, disrupt electron transport, and affect the activity of key enzymes, as well as the ultimately altering photosynthetic efficiency, impacting crop growth and yield. Thus, investigating the movement of NPs within plants and their effects on oxidative stress and photosynthesis is essential for understanding their biological effects. This knowledge also supports the assessment of the safety and potential applications of nanotechnology in agriculture.

#### 2.2.1. Transport of NPs in Plants

Plants can absorb NPs through three main pathways: seed infusion, foliar spraying, and root adsorption [67,68]. Numerous studies have shown that the size of NPs is crucial for their direct absorption by plants, as it influences their ability to enter through cell wall pores, stomata [69], membranes, and intercellular spaces [67]. From a chemical perspective, the surfaces of plant cell walls and phospholipid membranes carry a negative charge. Consequently, the surface charge properties of NPs determine the strength of electrostatic interactions with the cell interface, which affects both adhesion efficiency and the rate of internalization [70]. Therefore, the physical and chemical properties of NPs, such as size, surface charge, hydrophobicity, and chemical composition, collectively regulate their behavior across membranes and their subsequent biological effects [71].

There are three primary ways for NPs to enter cells (Figure 1): (1) Passive diffusion: NPs can directly penetrate the plasma membrane due to their small size. This process is influenced by several factors, including the size of the NPs, their hydrophobicity, elemental composition, charge, and geometry [72]. (2) Endocytosis: NPs can be actively transported into cells via endocytic pathways through a process known as endocytosis [73]. (3) Transmembrane transport: NPs can facilitate their own entry into plant cells by using transmembrane transporters or by regulating transmembrane channels [74].

The transport of NPs from cells to tissues in plants primarily occurs through either apoplastic or symplastic pathways. The pathway used depends on whether the NPs enter the plant through leaves/branches or roots [75]. When NPs are absorbed by the roots, they typically pass through the extracellular matrix and cell wall using the apoplastic pathway. From there, NPs enter the xylem vessels and move upward to vascular tissues and aboveground organs. It is important to note that once absorbed by the roots, NPs are mainly transported upward, with minimal transport occurring downward. Research has shown that the transport of NPs can be classified into top-down and bottom-up pathways.

The top-down pathway involves foliar-sprayed NPs penetrating the leaves through the stomata or epidermis. Once inside, they can be transported downward or bidirectionally through the phloem and xylem [76]. In contrast, the bottom-up pathway allows the roots to absorb NPs from the environment. These NPs then migrate to the stele through the endodermis or apoplast, and subsequently move to the stems and leaves via the xylem.

The symplast pathway requires NPs to overcome two significant barriers. First, they must penetrate the cell membrane to enter the cytoplasm, and second, they need to achieve intercellular transmission through plasmodesmata. In contrast, the apoplastic pathway allows NPs to travel along a continuous network of cell walls until they reach the Casparian strip in the endodermis. At this point, some NPs can be redirected into the symplast for further inward transport [77]. The apoplastic pathway is recognized as another major transport route. In this pathway, NPs are typically absorbed by the endoderm of the root system and then transported to the stems, leaves, and fruits of the plant via the xylem or phloem [78,79]. Researchers have investigated the mechanisms behind NP absorption by plant roots and found that the intercellular symplast pathway is the primary route for transport.

In contrast, foliar spray NPs are transported through the phloem and accumulate in various plant organs. When NPs are sprayed on leaves, they are primarily transported over long distances via the phloem. Initially, these NPs are retained in the leaf cuticle, some of them penetrate through the stomatal openings, and eventually, they rely on phloem sieve tubes to move towards the roots [80]. Although stomata represent only about 5% of the leaf’s surface area and their openings can be influenced by environmental factors such as carbon dioxide levels, humidity, and light intensity, their relatively large pore size, averaging around 10 μm, still allows for an effective entry point for NPs.

Symplastic transport depends on plasmodesmata to facilitate the exchange of materials between cells, allowing for short-distance transmission. Meanwhile, the xylem and phloem are responsible for long-distance transport, with the xylem leading in upward movement and the phloem in downward movement. The path that NPs take within the plant is closely linked to their location, physicochemical properties, and the physiological state of the plant. Once NPs enter the plant, they can have a significant impact on its physiological condition.

Symplastic transport relies on plasmodesmata to achieve intercellular material exchange and undertakes the short-distance transmission function between cells [81]. The xylem and phloem lead in the upward and downward long-distance transport, respectively [67,82]. It can be seen that the migration path of NPs is closely related to the location, physicochemical properties, and physiological state of plants. After NPs enter the plant, they will have a certain impact on the physiological state of the plant.

#### 2.2.2. Effects of NP Delivery Route on Plant Response

NPs, whether applied via foliar spraying or soil amendment, profoundly affect uptake efficiency, translocation pathways, and the downstream biological responses of plants. Clarifying how these two delivery modes differentially shape plant responses is therefore essential for optimizing practical strategies in nano-enabled agriculture.

Foliar spraying delivers NPs directly to the leaf surface, where they can penetrate via stomata, micro-cracks in the cuticle, or trichomes. This route is especially useful for rapid nutrient supplementation or for mitigating acute abiotic stress. For example, foliar application of ZnO NPs was shown to markedly raise photosynthetic pigment levels and antioxidant enzyme activities in wheat, thereby enhancing drought tolerance [20]. Likewise, Kohatsu et al. [19] reported that CuO NPs sprayed on lettuce significantly improved non-enzymatic antioxidant capacity and photosynthetic performance under stress. Once inside, foliar-applied NPs can undergo systemic transport through the phloem, although the efficiency of this process is modulated by particle size, surface charge, and the physicochemical properties of the leaf surface [64].

Soil application, in contrast, relies on root uptake, with NPs entering the plant predominantly through apoplastic and symplastic pathways. This mode is better suited for sustained nutrient provision or for manipulating rhizosphere stresses. Li et al. [49] found that Fe_3_O_4_ NPs added to soil promoted nutrient acquisition in rice by improving root architecture and increasing iron bioavailability. However, the complexity of the soil matrix—organic-matter adsorption, clay aggregation, and other factors—can reduce NP bioavailability, especially for metal oxide nanomaterials [77]. Soil pH, moisture, and microbial activity further influence NP speciation, stability, and ultimately their biological effects [45].

Critically, the same NP can elicit distinct physiological outcomes depending on how it is delivered. Servin et al. [44] observed that TiO_2_ NPs enhanced photosynthetic efficiency in cucumber when foliarly sprayed, whereas root application mainly altered root morphology with limited effects on the shoot. Similarly, Lian et al. [20] noted that soil-applied ZnO NPs were more effective at boosting Zn accumulation in rice grains, whereas foliar delivery preferentially elevated leaf Zn content and photosynthetic performance. Collectively, these findings underscore that the choice of application route should be tailored to the target tissue and the desired agronomic outcome.

#### 2.2.3. Oxidative Stress and Antioxidant System Response

Oxidative stress is a condition that occurs when the organism produces an excessive amount of free radicals, including reactive oxygen species (ROS) and reactive nitrogen species (RNS), in response to harmful stimuli. This overproduction can lead to damage in cells, tissues, and organs. To combat this issue, the organisms have developed a complex antioxidant defense system. This system works to eliminate excess free radicals, repair oxidative damage, and maintain redox balance.

After nanoparticles (NPs) enter the plant body, they induce the generation of reactive oxygen species (ROS) due to their high specific surface area and abundant surface-active sites. Metal oxide nanoparticles, such as titanium dioxide (TiO_2_) and zinc oxide (ZnO), can absorb photon energy and generate electron–hole pairs when exposed to UV irradiation [83]. These electrons and holes react with water molecules and oxygen adsorbed on the surface of the particles. This process can produce various ROS, including superoxide anion radicals (O_2_^−^·) and hydroxyl radicals (·OH) [84]. ROS can act as signaling molecules that activate the plant’s antioxidant defense mechanisms to maintain redox balance [85,86]. However, ROS are highly reactive and can attack biological macromolecules such as cell membranes, proteins, and DNA, leading to abnormal cell function and even cell death. Under uncontrollable abiotic stress, plants may produce excessive ROS, resulting in redox imbalance, cell damage, and hindered growth. On the other hand, NPs can induce ROS production in a controlled manner, stimulating the plant’s antioxidant defense mechanisms. This stimulation can lead to an increase in antioxidant enzymes, such as superoxide dismutase (SOD), catalase (CAT), and peroxidase (POD) [87]. These enzymes neutralize ROS, helping prevent cell damage [88]. For instance, silver nanoparticles (AgNPs) have been shown to induce ROS production in plant tissues, enhancing the antioxidant response and increasing plant resistance to various stressors [89]. Additionally, nanoparticles can promote the activity of antioxidant enzymes, enabling plants to effectively resist oxidative stress. For example, in wheat plants treated with copper oxide (CuO) nanoparticles synthesized from neem leaf extract, the activities of SOD, POD, and CAT increased, resulting in enhanced tolerance to oxidative stress induced by heavy metals like cadmium [90].

Nanoparticles not only affect antioxidant enzymes but also enhance non-enzymatic antioxidant compounds, improving plants’ ability to resist oxidative stress [91]. For instance, pretreating maize seeds with selenium nanoparticles (Se NPs) can activate the ascorbic acid–glutathione (AsA-GSH) cycle in vivo. This treatment resulted in a 35.4% increase in the level of ascorbic acid (AsA), while the content of reduced glutathione (GSH) also significantly increased by 37% [92]. The accumulation of these non-enzymatic antioxidants effectively boosts the plants’ resistance to oxidative stress [92]. Similarly, when *Atropa belladonna* is treated with manganese oxide nanoparticles (MnO_2_ NPs), there is a significant increase in non-enzymatic antioxidants such as total phenols and flavonoids. The accumulation of these substances further enhances the plant’s resistance to oxidative stress [93].

In summary, nanomaterials can induce controllable reactive oxygen species (ROS) bursts in plants. This process can simultaneously increase the activity of antioxidant enzymes and the content of non-enzymatic antioxidants, thereby enhancing the antioxidant defense system of plants and ultimately significantly improving their tolerance to stress.

#### 2.2.4. Photosynthesis

Photosynthesis is the fundamental biological process that drives crop growth and yield formation [69]. Both land and aquatic plants convert light energy into chemical energy through this process. It can be divided into two main stages: the light reaction (energy conversion) and the dark reaction (carbon assimilation). In the light reaction, light energy is captured and used to produce adenosine triphosphate (ATP), reduced nicotinamide adenine dinucleotide phosphate (NADPH), and reducec ferredoxin [94]. During the dark reaction, carbon dioxide is fixed into organic compounds using the energy and reducing power provided by ATP and NADPH [95].

Ribulose-1,5-bisphosphate carboxylase/oxygenase (commonly known as Rubisco), an important enzyme in the photosynthesis process, is a vital enzyme involved in the carbon assimilation process. Its primary function is to catalyze the reaction between carbon dioxide (CO_2_) and ribulose-1,5-diphosphate (RuBP), ultimately producing 3-phosphoglycerate. Despite its crucial role in carbon fixation, Rubisco has a low catalytic efficiency and often serves as a rate-limiting factor in photosynthesis [96,97]. These recent studies have shown that NPs can effectively enhance photosynthesis.

NPs have the potential to enhance the activity of Rubisco. For example, treatment with cerium dioxide nanoparticles (CeO_2_ NPs) significantly improved both Rubisco activity and carbonic anhydrase (CA) activity in mustard plants, thereby effectively promoting the dark reactions of photosynthesis [70]. Similarly, after treatment with titanium dioxide nanoparticles (TiO_2_ NPs), there was a notable increase in Rubisco activity and NADPH content in tomato leaves, which in turn effectively boosted the net photosynthetic rate of the plants [98].

Furthermore, NPs can also influence the synthesis of photosynthetic pigments and regulate the efficiency of the photosynthetic electron transport chain. In photosynthetic organisms, chlorophyll (Chl) serves as the primary pigment responsible for capturing light energy and converting it into chemical energy. Numerous studies have shown that treatment with metal oxide nanoparticles (MONPs) can markedly increase the levels of chlorophyll a and b, as well as carotenoids in plants, which in turn enhances the abundance of light-harvesting complex II (LHCII) [95]. As a pigment–protein complex, LHCII is essential for transferring captured light energy to photosystem I (PSI) and photosystem II (PSII) [99]. This promotes photoreduction reactions and electron transfer, ultimately optimizing light energy utilization and enhancing photosynthesis. Furthermore, MONPs possess photocatalytic properties that can initiate redox reactions and facilitate charge transfer. They may also function as electron and proton carriers for PSI, PSII, ATP synthase, and plastoquinone, thereby playing a complementary role in boosting the activities of Rubisco and carbonic anhydrase [95]. The primary role of the photosynthetic electron transport chain is to convert light energy into active chemical energy.

Research has shown that embedding single-walled carbon nanotubes (SWCNTs) in thylakoid membranes can enhance light energy utilization efficiency, accelerate electron transfer, and promote carbon capture, ultimately improving overall photosynthetic activity [100]. Additionally, carbon quantum dots (CDs) can convert absorbed photons into electrons, providing supplemental electrons for the photosynthetic electron transport chain and enhancing plant photosynthesis [101]. In a study conducted by Li Wei et al., it was found that after coating chloroplasts with CDs, the production of adenosine triphosphate (ATP) by the chloroplasts increased by 2.8 times, and the electron transport rate rose by as much as 25%, leading to enhanced photosynthesis in plants [102].

In summary, NPs can significantly improve the photosynthetic performance of plants by enhancing the light energy capture ability of chloroplasts, promoting carbon dioxide fixation, and optimizing the efficiency of the photosynthetic electron transport chain.

## 3. Potential Risks of NPs in Applications

Nanotechnology is currently one of the top research priorities in many countries due to its enormous potential and economic impact. Although the use of NPs opens up new avenues for innovative and sustainable agriculture, its potential threat to plants, soil organisms, and human health should also be carefully considered before commercial application [103]. The accumulation of nanomaterials in the environment and food chain may pose a risk to human health.

### 3.1. Effects on the Soil Ecosystem

With the rapid development and widespread application of NPs, various nanomaterials (NMs) are entering the environment at an unprecedented rate. Soil, being a significant medium for these NMs, is increasingly impacted by their cumulative effects [104]. Earthworms, as key biological indicator species in soil ecosystems, play an essential role in improving soil structure, decomposing organic matter, and nutrient cycling [105]. Studies have indicated that titanium dioxide (TiO_2_) in soil can severely affect the survival and growth of adult earthworms [106]. Cañas et al. found that zinc oxide nanoparticles (ZnO NPs) have a more pronounced reproductive toxicity to earthworms compared to TiO_2_ NPs [107]. ZnO NPs can impair the reproductive functions of earthworms, resulting in damage to gonadal tissues and abnormal expression of genes related to reproduction [108]. Moreover, carbon-based nanomaterials, such as carbon nanotubes (CNTs) and graphene oxide (GO), have also been shown to negatively impact the reproductive capacity of earthworms [109]. These findings highlight that the effects of NPs on soil ecology exhibit significant material specificity. Different types of NMs disrupt the survival, growth, and reproduction of key soil organisms, such as earthworms, through distinct toxic mechanisms, thereby threatening the structural and functional stability of soil ecosystems (Table 3).

Nanomaterials have the potential to alter the structure and function of soil microbial communities. Among soil bacteria, ammonia-oxidizing bacteria, which are crucial for the nitrogen cycle, have been reported to be particularly sensitive to certain types of NPs. For example, studies have shown that cerium oxide nanoparticles can significantly reduce the abundance and activity of ammonia-oxidizing bacteria in agricultural soil, leading to impaired nitrification and potential disruption of nutrient dynamics [111,112]. Given the critical role of nitrification in nitrogen availability for plants, this specific sensitivity poses a considerable risk to soil health and fertility. Additionally, silver nanoparticles exhibit broad-spectrum antimicrobial activity, capable of inhibiting the growth of various bacteria and fungi in soil, thereby affecting nutrient transformation and cycling. These silver nanoparticles indirectly harm bacteria by releasing Ag^+^ and generating reactive oxygen species (ROS) [113]. Soil microorganisms play a crucial role in maintaining soil fertility and controlling plant diseases. Therefore, the interference of NPs with soil microbial communities may have detrimental effects on agricultural production.

NPs interact with the non-biological environment, which can influence their behavior and ecotoxicological potential [114,115]. Soil ecosystems are frequently exposed to a variety of chemical stresses, raising concerns about NPs acting as carriers for both organic and inorganic chemical stressors [116]. For instance, Schwab et al. [117] reported that the toxicity of the low-concentration herbicide diazinon was significantly increased in the presence of carbon-based nanoparticles [118]. Similarly, fullerene NPs heightened the acute toxicity of the insecticide fipronil. These findings suggest that NPs can enhance the bioavailability and ecological risk of toxic pollutants through a carrier effect, thus introducing more complex and hidden pollution threats to soil ecosystems.

### 3.2. Transmission and Accumulation in the Food Chain

Some studies have reported on the long-term effects of nanoparticles on plant bioaccumulation and exposure, which could impact the food chain. When NPs enter the environment, they may be transmitted, accumulated, or even biomagnified through the food chain [119,120]. As primary producers, plants can absorb NPs applied to the soil or sprayed on their leaves through their roots and foliage, respectively. Primary consumers of plants, such as insects and small mammals, ingest these nanomaterials during feeding. Research has confirmed that gold nanoparticles can transfer from tobacco plants (primary producers) to tobacco moths (primary consumers) using models of both organisms, demonstrating biomagnification. The biomagnification factors for gold nanoparticles with sizes of 5, 10, and 15 nm are 6.2, 11.6, and 9.6, respectively [121]. Some of the NPs ingested by primary consumers are digested and absorbed into their tissues and organs, while others may be excreted through feces. However, the excreted NPs can still participate in the cycling of the food chain.

In aquatic ecosystems, algae exposed to a suspension of TiO_2_ NPs over an extended period showed significant bioaccumulation. In the treatment group with a concentration of 100 mg/L, the TiO_2_ concentration in the algae cells reached 2.5 μg/g (dry weight). These NPs were then transferred to rotifers (consumers) through dietary exposure. Biomagnification occurred in this simplified food chain, with several biomagnification factors (BMF) exceeding 1, and a maximum BMF of 2.3 [122]. A similar phenomenon was observed in terrestrial ecosystems. After spraying metal-based nanoparticles (CeO_2_) and polymer-based nanoparticles (deuterated polystyrene, DPS) on the leaves of cherry radish, mixed exposure increased the absorption of cerium (Ce) by the plants but did not significantly affect the absorption of DPS. Notably, the migration of NPs from the ground to the roots remained unchanged despite the mixed exposure. The nutrient transfer efficiency of snails feeding on either the aboveground or root parts of the plants for Ce and DPS decreased, with a significant reduction in the transfer efficiency of Ce in root-feeding snails. Additionally, mixed exposure led to biomagnification of DPS in the digestive glands and soft tissues of the snails [123]. This indicates that mixed exposure not only affects the migration of NPs in plants but also alters their transport behavior within the food chain.

Previous studies have identified a terrestrial food chain model comprising kidney beans, Mexican bean beetles, and stingray bugs. When beans were planted in soil contaminated with CeO_2_ NPs, the concentration of cerium (Ce) in the roots reached 26 μg/g after 36 days, while the aboveground parts of the plants contained 1.02 μg/g of Ce. After Mexican bean beetles fed on the leaves that contained CeO_2_ NPs, the concentration of Ce in their tissues was higher than that found in their excreta. This pattern indicated biomagnification through the food chain—from plants to adult beetles and then to stingray bugs—with an amplification factor of 5.3. However, about 98% of the Ce was excreted by beetle larvae after consuming leaves containing CeO_2_ NPs [46]. This suggests that the transfer efficiency of NPs is significantly influenced by the digestion, absorption, and excretion mechanisms in animals.

In summary, the transmission behavior of nanomaterials in the food chain is highly complex and influenced by several factors, including their physical and chemical properties, the absorption pathways in plants, the feeding habits of animals, and their digestive and excretion mechanisms, as well as the conditions of single or mixed exposures. On the one hand, nanoparticles such as gold, TiO_2_, and CeO_2_ can accumulate in organisms and show biomagnification (BMF > 1). On the other hand, factors like mixed exposure and the physiological characteristics of animals can reduce transfer efficiency. The accumulation and transfer of NPs may pose potential threats to the physiological functions of organisms at all levels and affect the stability of ecosystems.

### 3.3. Potential Threats to Human Health

For human health, the enrichment and amplification of nanomaterials through the food chain may pose potential risks [124]. As consumers at the top of the food chain, humans may be indirectly exposed to NPs by ingesting crops, livestock and poultry products, and aquatic products contaminated by NPs [125]. Although the current research on the human health effects of NPs is not sufficient, there is evidence that nanomaterials may have toxic effects on cells, tissues, and organs after entering the human body. Their potential mechanisms include inducing oxidative stress and inflammatory response, thereby damaging cell structure and interfering with normal physiological functions. The main routes of human exposure to NPs include respiratory inhalation, direct eye or skin contact, and gastrointestinal ingestion [126,127]. NPs can further react with tissues, blood, and body fluids, and even penetrate the blood–brain barrier into the central nervous system (CNS), affecting the function of important organs such as the brain and heart [128]. In addition, some NPs may also interfere with the normal regulation of the endocrine system and immune system, and long-term exposure may increase the risk of chronic diseases.

Furthermore, it is crucial to encourage awareness and promote the ethical use of NPs within the agricultural sector. This involves developing and adhering to strict application guidelines, conducting comprehensive life-cycle assessments, and fostering responsible practices among stakeholders to minimize unintended environmental release. Such proactive measures are essential to prevent the contamination of diverse ecosystems and ensure the sustainable and safe integration of nanotechnology in agriculture.

## 4. Conclusions

Nanotechnology presents a promising yet complex tool for enhancing global food security. This review confirms that NPs can improve crop productivity and stress resilience by regulating key physiological processes. However, biphasic effects, complex mechanisms of action, and unresolved environmental risks hinder their translation from the laboratory to the field.

The most prominent feature of NP–plant interactions is the biphasic, concentration-dependent response. At optimal concentrations, NPs such as Ag, ZnO, and TiO_2_ act as effective plant nano-priming agents, improving seed germination vigor, root architecture, and biomass accumulation (Table 1). These promotive effects are primarily attributed to the unique properties conferred by their nanoscale size, including enhanced nutrient uptake, improved water retention, and targeted delivery of bioactive compounds [11,26]. For instance, the ability of NZVI to induce cell wall loosening [47] and the capacity of chitosan nanoparticles to stabilize auxin [15] demonstrate the complex physicochemical interactions underpinning growth promotion.

Conversely, the same ENMs can induce phytotoxicity beyond a critical threshold, manifested as oxidative burst, membrane damage, and growth inhibition. The suppression of root elongation by high doses of γ-Fe_2_O_3_ NPs [48] and the impairment of pollen tube development by GO [59] highlight the risks of improper dosage. This hormetic effect underscores a fundamental principle: the biological outcomes of NP applications are not determined solely by the material itself but emerge from its interactions with plant systems at specific concentrations. Therefore, defining a species- and environment-specific “therapeutic window” is crucial for safe application.

To exert their promoting or inhibiting effects, NPs must first successfully enter the plant body and reach their target sites. This process is fundamentally governed by their intrinsic physicochemical properties, such as size and surface charge. These properties determine whether NPs can effectively pass through plant barriers, such as the cell wall and stomata, and regulate their transmembrane behavior [66,68,70]. For instance, positively charged carbon dots (CDs) have been shown to exhibit enhanced barrier penetration capability due to their significant interactions with cell wall components [69]. After entering the plant, the migration pathway of NPs depends on their entry site: those absorbed by the roots primarily move via the apoplastic pathway toward the xylem and are transported upward (bottom-up) to the stems and leaves, while foliar-applied NPs are predominantly redistributed downward (top-down) through the phloem [74,75,79]. Therefore, the physicochemical attributes of NPs and their resulting distribution patterns within the plant serve as the logical foundation for understanding their subsequent biphasic biological effects.

The plant-stimulatory effects of NPs are mediated through a complex, interconnected network of physiological mechanisms, rather than isolated pathways. A core mechanism involves the modulation of redox homeostasis. Counter to the purely deleterious role of ROS, evidence indicates that NPs can induce a controlled, signaling-level ROS burst that “primes” the plant’s antioxidant defense system [84,85]. This nano-priming effect involves upregulating enzymatic antioxidants (SOD, CAT, POD) and enhancing the non-enzymatic antioxidant pool (e.g., the AsA-GSH cycle) [91], thereby improving the plant’s capacity to withstand subsequent abiotic stresses.

Concurrently, many NPs directly enhance photosynthetic efficiency through multiple routes: (i) increasing the abundance of light-harvesting complexes (LHCII) and chlorophyll content [85]; (ii) serving as artificial electron donors or facilitating electron transport, as observed with CDs and single-walled carbon nanotubes (SWCNTs) [99,101]; and (iii) upregulating the activity of key Calvin cycle enzymes, particularly Rubisco [69,101]. The synergistic effect of improved light reactions and enhanced carbon fixation provides a solid foundation for the observed increases in biomass and yield.

However, the potential for agricultural applications of NPs is tempered by significant environmental risks and ecotoxicological concerns. The “soil–plant–food chain” continuum represents a critical pathway for NP exposure and potential risk. Once introduced into the soil matrix, NPs may adversely affect earthworms and microbial communities. The toxicity of ZnO and carbon nanotubes to earthworms [106,108] and the inhibition of ammonia-oxidizing bacteria by CeO_2_ NPs [111,112] threaten the integrity of soil nutrient cycling and overall health. A particularly concealed risk is NPs acting as vectors for co-contaminants; studies showing that carbon NPs enhance the toxicity of pesticides like diuron [116] reveal a potential for composite stress that cannot be captured in single-contaminant risk assessments.

Regarding food safety, perhaps the most concerning evidence is for trophic transfer and biomagnification. The biomagnification of Au NPs in terrestrial food chains [120] and TiO_2_ NPs in aquatic models [121] challenges earlier assumptions that NPs remain inert or become diluted within ecological networks. These findings necessitate a precautionary approach, as long-term, low-level dietary exposure could pose unforeseen risks to human health, including inflammatory responses and organ-specific toxicity [125,126].

The application of nanotechnology in agriculture exhibits a distinct dual character: On one hand, NPs serve as efficient plant regulation tools by leveraging their unique physicochemical properties. They significantly enhance crop productivity and stress resistance through multiple pathways, including improved nutrient uptake, enhanced antioxidant defenses, and increased photosynthetic efficiency. On the other hand, their environmental fate and ecological risks remain major challenges, encompassing migration and transformation in soil–plant systems, potential impacts on biodiversity, and biomagnification along the food chain.

## 5. Future Directions

While NPs show great potential for enhancing crop stress tolerance and productivity, their transition from fundamental research to field application faces multiple challenges, including unclear mechanisms, undefined risks, and a lack of regulatory frameworks. Current research findings paint a promising yet incomplete picture. Translating this potential into safe and sustainable agricultural practices will require systematic breakthroughs across the following three dimensions in future studies.

### 5.1. Mechanism Exploration: From Phenomenon to Essence

Existing studies have preliminarily observed the significant effects of NPs on plant nutrient absorption and hormone signaling. For instance, ZnO NPs have been confirmed to act as an efficient zinc source, promoting nutrient accumulation in crops [56], while TiO_2_ NPs regulate root development by mimicking auxin activity [34]. However, the underlying mechanisms behind these phenomena—including the interactions between NPs and nutrient transport proteins, key regulatory nodes in hormone signaling pathways, and potential epigenetic influences—remain to be systematically elucidated. Future research should integrate multi-omics technologies and molecular biology approaches to unravel the molecular basis of NP-mediated plant physiological regulation at the levels of gene expression, protein function, and metabolic networks, thereby providing a theoretical foundation for precision agricultural applications.

### 5.2. Problem-Oriented Approach: From Understanding to Control

Current research has identified three core challenges for nano-agricultural technology: first, the absorption, transport, and metabolic mechanisms of NPs in different crops are not yet fully understood; second, the phytotoxicity of high-concentration NPs and their long-term ecological risks in soil require urgent assessment; and third, the potential threats of NPs to human health through food chain transmission need to be quantified. To address these issues, future studies should establish an integrated “mechanism–effect–risk” research framework, develop safety application threshold systems through long-term exposure experiments, and focus on investigating the combined effects of NPs with other environmental pollutants.

### 5.3. Application Expansion: From Basic Research to Systemic Solutions

While advancing fundamental research, simultaneous efforts must be made in technological innovation and management at the application level. This includes developing smart delivery systems capable of responding to environmental signals to achieve precise release and efficient utilization of NPs; constructing a comprehensive regulatory standard framework covering the entire “synthesis–application–monitoring–fate” chain; and establishing big data-based risk early warning and control platforms. Furthermore, interdisciplinary collaboration should be strengthened, fostering deep cooperation among nanomaterial scientists, agronomists, ecologists, food safety experts, and policymakers. Together, these efforts will build safety barriers for nano-agricultural technology from the laboratory to the field, and from the field to the table, ultimately ensuring both food security and ecological safety.

## Figures and Tables

**Figure 1 cimb-47-01024-f001:**
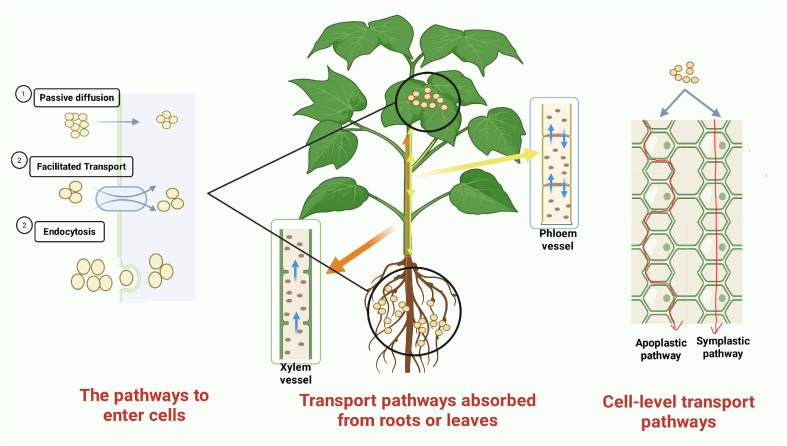
The figure shows that nanoparticles (NPs) enter cells through three mechanisms: passive transport, facilitated diffusion, and endocytosis. NPs entering via the roots are mainly transported upward through the xylem, whereas those entering via the leaves are primarily transported through the phloem in a biphasic manner. Within plants, NPs are transported via two pathways: the apoplastic pathway and the symplastic pathway. In the symplastic pathway, NPs pass through the cell membrane into the cytoplasm and then move through plasmodesmata to adjacent cells. In the apoplastic pathway, NPs move through extracellular spaces, such as the cell wall and intercellular space, forming a continuous ‘free space’ network.

**Table 1 cimb-47-01024-t001:** Effects of different nanomaterials on seed germination, vegetative growth, and reproductive growth of plants.

Nanomaterial	Concentration	Size (nm)	Target Plant Species	Treatment Methods	Action Period	Effect	Reference
Ag NPs	10 mg/L, 80 mg/L,	28.32; 29	Watermelon (*Citrullus lanatus*)	soaking treatment	seed germination stage	Improve the seed germination rate and soluble sugar content of seedlings	Acharya et al. (2020); [11]
Ag NPs; Ti NPs	5 mg/mL; 100 mL; 10 mg/100 mL; 15 mg/100 mL	39 21	Corn (*Zea mays*); Rice (*Oryza sativa*)	soaking treatment	seed germination stage	Significantly increased the germination rate of rice and corn seeds.	Iqbal et al. (2021); [12]
ZnO NPs; Fe NPs		20–30	Wheat (*Triticum aestivum* L.)	soaking treatment	vegetative stage	Increasing plant height and dry weight had a positive effect on the photosynthesis of wheat.	Rizwan et al. (2019); [13]
Ag NPS	10 mg/L	10–35	Wheat (*Triticum aestivum* L.)		vegetative stage	The early growth of seedlings was effectively promoted by reducing ROS toxicity.	Kannaujia et al. (2019); [14]
CSNPs	5 μg/ml	80–200	Wheat (*Triticum aestivum* L.)	irrigation treatment	vegetative stage	Promote the increase in IAA concentration in wheat buds and roots, thereby promoting the growth of wheat seedlings.	Li et et al. (2019); [15]
MgO NPs	100 mg/L	12	green gram (*Vigna radiata* L.)	soaking treatment	seed germination stage	Improve seed germination rate and seedling vigor.	Anand et al. (2020); [16]
Fullerol	50 mg/L		Wheat (*Triticum aestivum* L.)	soaking treatment	seed germination stage	Improving seed germination under drought stress.	Kong et al. (2023); [17]
Nanoscale zero-valentcobalt (NZVC)	0.17 mg/kg	40–60	Soybean (*Glycine max* L.)	soaking treatment	vegetative stage	Improve growth parameters such as plant height, leaf area, and stem and leaf dry weight; it has a positive effect on the growth of plants in the vegetative growth period.	Hong et al. (2019); [18]
CuO NPs	20 mg per plant		Lettuce (*Lactuca sativa*)	foliage spray	vegetative stage	Enhance non-enzymatic antioxidant activity, improve plant stress tolerance, and increase plant photosynthetic pigment content.	Kohatsu et al. (2021); [19]
ZnO NPs	100 mg/L		Wheat (*Triticum aestivum* L.)	foliage spray	reproduction stage	Increase wheat grain, increase yield, and significantly increase the nutrient content of wheat seeds.	Lian et al. (2024); [20]
MnFe_2_O_4_ NMs	10 mg/L		Tomato (*Solanum tuberosum*)	foliage spray	reproduction stage	It increased pollen activity and egg size, and improved plant seed setting rate.	Yue et al. (2022); [21]
CeO_2_ NMs	10 mg/kg		Cucumber (*Cucumis sativus* L.)	irrigation	reproduction stage	It can induce early flowering and improve fruit yield and quality.	Feng et al. (2023); [22]
Se Engineered Nanomaterials (ENMs)	75 μg/kg		Cherry tomato (*Solanum lycopersicum var. cerasiforme*)		reproduction stage	Improve the yield and quality of tomatoes	Cheng et al. (2022); [23]
Se NPs	50, 100 and 200 mg/mL		Faba Bean (*Vicia faba*)	soaking treatment	reproduction stage	The dry weight per bean, the number of seeds per plant, and the number of pods per plant increased.	Soliman et al. (2024); [24]

**Table 2 cimb-47-01024-t002:** Effects of NPs on plant pollen vitality, pollen tube growth and reproductive development.

Nanomaterial	Concentration	Plant Species	Effect on Pollen/Pollen Tube	Effect on Reproduction	Reference
ZnO NPs	10–100 mg/L	Lily (*Lilium* spp.)	Inhibited pollen germination and tube elongation; no effect on pollen viability.	Potential negative impact on fertilization.	Yoshihara et al., 2021; [59]
Graphene Oxide (GO)	50–200 mg/L	*Nicotiana tabacum*, *Corylus avellana*	Significant inhibition of pollen germination and tube growth; abnormal tube morphology.	Impaired sexual reproduction.	Candotto Carniel et al., 2018; [60]
Ag NPs	10–50 mg/L (foliar spray)	Peach (*Prunus persica*)	Increased pollen grain number and viability; no malformation.	Enhanced pollination and fertilization efficiency.	Mosa et al., 2021; [61]
MnFe_2_O_4_ NMs	10 mg/L (foliar spray)	Tomato (*Solanum lycopersicum*)	Increased pollen activity.	Improved seed setting rate.	Yue et al., 2022; [21]
CeO_2_ NPs	10 mg/kg (soil)	Cucumber (*Cucumis sativus*)	–	Induced early flowering and improved fruit yield and quality.	Feng et al., 2023; [22]
Se ENM	75 μg/kg (soil)	Cherry tomato (*Solanum lycopersicum* var. *cerasiforme*)	–	Increased flower size, improved fruit yield and nutritional quality.	Cheng et al., 2022; [23]
Fullerol (C_60_(OH)_20_)	50 mg/L (seed priming)	Bitter melon (*Momordica charantia*)	–	Increased fruit number (59%), individual fruit weight (70%), total yield (128%), and bioactive compounds.	Kole et al., 2013; [62]

**Table 3 cimb-47-01024-t003:** Effects of different nanomaterials on earthworms.

Nanomaterial	Concentration	Earthworm Species	Exposure Duration	Key Effects	Reference
ZnO NPs	100–1000 mg/kg	Eisenia fetida	28 days	Reproductive toxicity; damaged gonadal tissues; abnormal gene expression	Cañas et al. (2011); [17]
TiO_2_ NPs	1000–2000 mg/kg	Eudrilus euginiae	14 days	Reduced survival; growth inhibition; oxidative stress	Priyanka et al. (2018); [106]
CNTs	500–1000 mg/kg	Eisenia fetida	28 days	Impaired reproduction; histopathological damage to body wall	Duo et al. (2022); [109]
Graphene Oxide	100–500 mg/kg	Eisenia fetida	14 days	Reduced reproductive output; oxidative damage; inflammation response	Duo et al. (2022); [109]
Ag NPs	10–100 mg/kg	Eisenia fetida	14 days	Mortality at high doses; reduced growth and reproduction	Shoults-Wilson et al. (2011); [110]

## Data Availability

No new data were created or analyzed in this study. Data sharing is not applicable to this article.

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
