# Peer review of "Effects of Nanomaterials on Crops"

_cimb, 2025, doi:10.3390/cimb47121024_

Round 1
Reviewer 1 Report
Comments and Suggestions for Authors
Generally speaking, it is a very interesting article that, in addition to providing valuable information, offers a profound reflection on the different areas that must be studied and monitored regarding the use of nanomaterials in the agricultural field.
Below, the following points are added, which could improve the manuscript:
– Improve image quality.
– Review the English and the formatting throughout the manuscript. There are double spaces in several parts of the document. Also, check the numbering of the sections.
Some of the comments on the document arise from the journal’s scope in molecular biology.
– Section 2.1.1. The authors state that NPs can adhere to or penetrate the seed coat, but this is still not entirely clear. It is necessary to explain further and develop this mechanism to understand the origin of the NP effect better, for example, from a biochemical and signaling perspective.
– In this same section, it is important to mention and more clearly develop the effect of high concentrations when using Ag and ZnO NPs, as well as the positive effects of low concentrations of carbon allotropes on germination.
– The interaction mechanisms of AgNPs, ZnONPs, or even carbon allotropes with seeds or plant models are very different. If the manuscript discusses the effects of these materials, then it is also necessary to mention and describe these mechanisms.
– Section 2.2. Several studies in the literature report the effect of applying or not NPs in crops; therefore, it is important to review and further elaborate on the mechanisms by which nanomaterials increase root length and diversity.
– Section 2.2.1. It is important to add a subsection addressing the effect of the NP delivery route in plants; that is, do NPs have the same effect when applied via foliar spray or through the soil?
– Section 2.1.3. The work related to pollen and its effect on plant reproduction is very interesting and novel. It would be useful to add a table illustrating the use of different materials on these two variables (pollen and reproductive effects).
– Section 2.2.1. It is recommended to include a representative diagram to visually illustrate the transport of nanomaterials in plants and within cells.
– Section 3.1. The effect of NPs on soil is a topic of great importance. It is advisable to include a table showing the effect of different NPs on earthworms as a function of concentration.
It is also important to determine whether any agricultural soil bacterial community has been reported to be more sensitive to a particular type of nanoparticle.
– Section 3.3. It is considered important to add a paragraph encouraging awareness in humans about the ethical use of NPs in the agricultural sector to avoid contamination of different ecosystems.
Comments on the Quality of English Language
Review the English and the formatting throughout the manuscript. There are double spaces in several parts of the document. Also, check the numbering of the sections.
Author Response
General Comment
Generally speaking, it is a very interesting article that, in addition to providing valuable information, offers a profound reflection on the different areas that must be studied and monitored regarding the use of nanomaterials in the agricultural field.
Below, the following points are added, which could improve the manuscript:
– Improve image quality.
– Review the English and the formatting throughout the manuscript. There are double spaces in several parts of the document. Also, check the numbering of the sections.
Answer: We sincerely thank the reviewer for the constructive comments and recognition of our study's structure and relevance. We fully acknowledge that this work represents an important initial step in understanding nanoparticle effects on plant, and we have revised the manuscript to improve the image quality and our manuscript.
Specific Comments
Some of the comments on the document arise from the journal’s scope in molecular biology.
Question 1.– Section 2.1.1. The authors state that NPs can adhere to or penetrate the seed coat, but this is still not entirely clear. It is necessary to explain further and develop this mechanism to understand the origin of the NP effect better, for example, from a biochemical and signaling perspective.
Answer: Thank for your good comments, we have added the mechanisms through which nanomaterials affect seeds, pleased find in line 86-101.
Question 2.– In this same section, it is important to mention and more clearly develop the effect of high concentrations when using Ag and ZnO NPs, as well as the positive effects of low concentrations of carbon allotropes on germination.
Answer: Thank for your good comments, we have added the effects of high-concentration silver/zinc oxide nanoparticles and the positive effects of low-concentration carbon allotropes on germination, pleased find in line 131-136 and 125-130.
Question 3.– The interaction mechanisms of AgNPs, ZnONPs, or even carbon allotropes with seeds or plant models are very different. If the manuscript discusses the effects of these materials, then it is also necessary to mention and describe these mechanisms.
Answer: Thank for your good comments, we have added details on the interactions of AgNPs, ZnONPs, or carbon allotropes with the seed or plant models, pleased find in line 86-102 and 114-130.
Question 4.– Section 2.2. Several studies in the literature report the effect of applying or not NPs in crops; therefore, it is important to review and further elaborate on the mechanisms by which nanomaterials increase root length and diversity.
Answer: Thank for your good comments, following your suggestion, we have positioned this section as 2.1.2 and revised it accordingly to elucidate the mechanisms by which nanomaterials influence root length and diversity, pleased find in line 152-181.
Question 5.– Section 2.2.1. It is important to add a subsection addressing the effect of the NP delivery route in plants; that is, do NPs have the same effect when applied via foliar spray or through the soil?
Answer: Thank for your good comments, we have added a subsection after 2.2.1 to illustrate the influence of the transport pathways of nanoparticles in plants: the effects of nanoparticle application through foliar spraying or through soil application are different, pleased find in line 402-433.
Question 6.– Section 2.1.3. The work related to pollen and its effect on plant reproduction is very interesting and novel. It would be useful to add a table illustrating the use of different materials on these two variables (pollen and reproductive effects).
Answer: Thank for your good comments, we have added a table showing the impact of nanomaterials on plant pollen and reproductive effects, pleased find in Table 2 in our revised manuscript.
Question 7.– Section 2.2.1. It is recommended to include a representative diagram to visually illustrate the transport of nanomaterials in plants and within cells.
Answer: Thank for your good comments, we have added a schematic diagram of the transport of nanoparticles in plants as shown in Figure 1, pleased find in Figure 1 in our revised manuscript.
Question 8.– Section 3.1. The effect of NPs on soil is a topic of great importance. It is advisable to include a table showing the effect of different NPs on earthworms as a function of concentration.
Answer: Thank for your good comments, we have added a table showing the impact of nanomaterials on earthworms, pleased find in Table 3 in our revised manuscript.
Question 9. It is also important to determine whether any agricultural soil bacterial community has been reported to be more sensitive to a particular type of nanoparticle.
Answer: Thank for your good comments, we have confirmed that there are reports indicating that any agricultural soil bacterial community is more sensitive to specific types of nanoparticles, and have added new literature support, pleased find in line 563-573.
Question 10.– Section 3.3. It is considered important to add a paragraph encouraging awareness in humans about the ethical use of NPs in the agricultural sector to avoid contamination of different ecosystems.
Answer: Thank for your good comments, we have already added a paragraph to encourage human awareness of the importance of ethical use of nanoparticles in the agricultural sector and have proposed relevant measures, pleased find in line 655-661.
Reviewer 2 Report
Comments and Suggestions for Authors
Dear Editor, MDPI Journal, thank you for selecting me to review this important Chapter (Effects of nanomaterials on the growth of crops)
There are some queries and corrections
All parts were written well and in simple compounds
Graphical abstract may be transport to the end of the abstract section
What the different between nutrition stage and vegetative stage (In page 5)
I prefer the Vegetative stage
In this paragraph (In addition, the germination rate (89.5%) and single seed dry weight (0.0198 g) of soybean seeds were significantly increased treated with 1 g/L of ZnO nanoparticles under drought conditions [30]), I think that, the percentage of single seed dry weight is the best.
The second part (Effects of nanomaterials on the growth and development of crops)
of the chapter need more survey on different nanoparticles and different crops may be need added another studies as https://doi.org/10.1038/s41598-024-77353-2
Author Response
Response to Reviewer #2
There are some queries and corrections
Specific Comments
Question 1. All parts were written well and in simple compounds
Answer: Thank for your good comments, we have carefully checked our revised manuscript, we have revised the manuscript to improve the image quality and our manuscript.
Question 2. Graphical abstract may be transport to the end of the abstract section
Answer: Thank for your good comments, we have already made changes to the graphics, pleased find in line 30-31.
Question 3. What the different between nutrition stage and vegetative stage (In page 5), I prefer the Vegetative stage
Answer: Thank for your good comments, we have replaced the nutrition stage with the vegetative stage, pleased find in line 146-147.
Question 4. In this paragraph (In addition, the germination rate (89.5%) and single seed dry weight (0.0198 g) of soybean seeds were significantly increased treated with 1 g/L of ZnO nanoparticles under drought conditions [30], I think that, the percentage of single seed dry weight is the best.
Answer: Thank for your good comments, since we couldn't find the proportion of dry weight increase, we replaced this example, pleased find in line 111-124.
Question 5. The second part (Effects of nanomaterials on the growth and development of crops) of the chapter need more survey on different nanoparticles and different crops may be need added another studies as https://doi.org/10.1038/s41598-024-77353-2
Answer: Thank for your good comments, we have added the content of this study to the table, pleased find in line 81-83.
Reviewer 3 Report
Comments and Suggestions for Authors
See attached file

Author Response
Response to Reviewer #3
General Comment
- Summary
The increase in the world population and the consequent need to expand agricultural production have often led to the excessive use of chemical fertilizers and other agrochemicals. In this context, the application of nanomaterials as a tool to enhance crop productivity within sustainable and innovative agricultural systems has emerged as one of the most prominent research areas in recent years. However, biosafety concerns associated with nanoparticles, including their environmental impacts and potential transfer through the food chain, may pose significant risks to human health.
Answer: We sincerely thank the reviewer for the constructive comments and recognition of our study's structure and relevance. We fully acknowledge that this work represents an important initial step in understanding nanoparticle effects on plant, and we have revised the manuscript to accurately reflect the scope and limitations of our findings.
Question 1. Overall, the manuscript is well written and does not present major flaws; however, certain aspects require improvement, as noted in the following section, “Specific Comments by Section”. The title clearly reflects the contents, and appropriate keywords were provided. The manuscript is well organized, appropriately sized, and properly referenced, though minor improvements are suggested (see below).
Answer: We sincerely thank for the constructive comments and recognition of our study's structure and relevance. We have carefully modified our manuscript.
Question 2. On page 2, a suitable abbreviation for nanoparticles (NPs) is correctly introduced. However, throughout most of the manuscript, this abbreviation is not used, and the full term appears repeatedly. I suggest that wherever the word ‘nanoparticles’ appears in the document, it be replaced with the corresponding abbreviation, NPs.
Answer: We sincerely thank for your good comments, we have carefully checked all the abbreviation in our revised manuscript, and have modified the “nanoparticles” to “NPs”.
Question 3. The abbreviation “et al.” should consistently appear in italics throughout the manuscript.
Answer: We sincerely thank for your good comments, we have carefully checked all the abbreviation“et al.”in our revised manuscript, and have modified them in italics.
Specific Comments
Question 4. Abstract:“Finally, this paper emphasizes the need for the enhancement of long-term ecological security assessments and the developing of intelligent delivery systems in future research, which can ensure the safe and efficient application of nanoagricultural technologies.”I suggest that the underlined word be replaced with review.
Answer: We sincerely thank for your good comments, we have revised the “paper” to “review” in our abstract, please find in line 24-27.
Question 5. 1. Introduction: Page 1: “The world population is expected to reach 9.6 billion, and necessitating a 70-100% increase in agricultural production to fulfill escalating food requirements by 2050 [1,2].” • Recent references should be cited, as the current ones are from 2017 and 2012, respectively. According to some authors, these predictions are no longer the most accurate. The reference provided below specifically addresses this controversy.
David Adam, 2021. How far will global population rise? Researchers can’t agree.
Nature, 597(7877), 462-465.
Answer: Thank for your good comments, we have updated the two references in our introduction, pleased find in reference 1-2.
Question 6. Page 2:“Nanomaterials are the organic, inorganic or hybrid materials ranging in size from 1 to 100 nm, possessing unique physical, chemical, and biological properties, such as high specific surface area, strong adsorption capacity, and adjustable surface charge.”• I suggest including the highlighted words.
Answer: Thank for your good comments, we have just included the highlighted words, pleased find in line 45-47.
Question 7. “Consequently, these nanoparticles NPs can accumulate at specific sites within plants and exhibit distinct biological functions [6, 7].” • Shouldn’t reference 6 also be cited here? • Include the yellow highlighted word and delete the unnecessary content (highlighted in red).
Answer: Thank for your good comments, we have deleted the reference 6 and included the highlighted words, pleased find in line 50-51.
Question 8. Grafical Abstract: • Is this representation a graphical abstract or a figure? If it is a graphical abstract, shouldn’t it appear before the introduction? If it is a figure, it should have a caption.
Answer: Thank for your good comments, it is a graphical abstract, and it was updated before the introduction, pleased find it in our revised manuscript.
Question 9. Relevant Sections: • The title of this section seems inappropriate; I recommend changing it to “Effects of Nanomaterials on Plants”.
Answer: Thank for your good comments, we have changed our title to “Effects of Nanomaterials on Plants”, pleased find it in our revised manuscript.
Question 10.Pages 3 - 4: Table 1: The table is not easy to read, as the separation of content in the ‘Effect’ column between the different rows for each nanomaterial is unclear; that is, the rows should have more spacing to make the separation apparent. Attention should also be paid to the spacing between columns, which is sometimes inadequate. The authors should review certain formatting aspects, including missing spaces between words or numbers, missing dots, or extra dots.
Other aspects to consider:
Header: - All words should begin with a capital letter. - column “Target Plant Species”. Include the highlighted word. - column “Effect”. Correction of the word. Data section: - column “Nanomaterial”: Row 8: include the designation NZVC; Nanoscale zero-valent cobalt (NZVC). Row 13: include the designation ENM;Engineered Nanomaterials (ENMs) ; - column “Target Plant Species”: Include the scientific names of the species under study. - column “Reference”: et al. It should be in italics and end with a dot. - column “Target Plant Species”: Last row: the species should be corrected to cherry tomatoes. - column “Effect”: Last row: I suggest that cherries be replaced with tomatoes.
Answer: Thank for your good comments, we have carefully modified all the details in the Table 1 follow all the comments, pleased find it in our revised manuscript.
Question 11.“Seed germination is the critical starting point for the growth of higher plants, which is particularly sensitive to environmental stresses. Adverse conditions can lead to a lower germination rate and seedling activity. Researchs indicates that treatments of various nanoparticles can significantly enhance the germination performance and physiological activity of crop seeds under different stress.” • Include the yellow highlighted letters and delete the unnecessary content (highlighted in red).
Answer: Thank for your good comments, we have carefully modified them, pleased find it in our revised manuscript in line 102--106.
Question 12.“Almutairi’s work also shows the treatments with Ag NPs were significantly higher on the germination rate, seedling length, fresh weight, and dry weight of tomato seeds than the control group when exposed to salt stress [28].”• Include the yellow highlighted letters.
Answer: Thank for your good comments, we have carefully modified them, pleased find it in our revised manuscript in line 108-111.
Question 13.“In summary, the suitable concentration of nanomaterials can promote the germination of plant seeds and enhance the resistance to the biotic and abiotic stresses. Howevere,…”• Delete the unnecessary content (highlighted in red)
Answer: Thank for your good comments, we have carefully modified it, pleased find it in line 143.
Question 14. Page 5:“For instance, the model plants like Arabidopsis thaliana and economic crops such as cucumber, titanium dioxide nanoparticles (TiO2NPs) can promote the elongationof the primary root, and also stimulate the formation of lateral roots [34, 35],.”• Delete the unnecessary content (highlighted in red).
Answer: Thank for your good comments, we have modified the sentences, pleased find it in line 177-181.
Question 15. “Nanoscale zero-valent iron (NZVI) promotes root extension by inducing the relaxation of cell walls through hydroxyl radicals (.OH), which in turn boosts root metabolism activity and enhances nutrient uptake [38].”• Include the yellow highlighted letters. The radical dot of the hydroxyl group should be placed above the OH group.
Answer: Thank for your good comments, we have included the yellow highlighted letters, and the radical dot of the hydroxyl group also be placed above the OH group. , pleased find it in line 169-173.
Question 16. “Similarly, Li et al. found that 20 mg/L γ-Fe2O3 NPs significantly enhanced corn root length growth by 11.5%. In contrast, higher concentrations of 50 mg/L and 100 mg/L γ-Fe2O3 NPs reduced root length by 13.5% and 12.5%, respectively [42].”• Include the yellow highlighted words and delete the unnecessary content (highlighted in red).
Answer: Thank for your good comments, we have included the yellow highlighted letters, and deleted the unnecessary content, pleased find it in line 185-188.
Question 17. “Furthermore, treatment with 500 mg/kg of FeO NPs significantly improve assimilation rates and intercellular concentrations of CO2, as well as transpiration rates, and…”• Ensure that the number highlighted in yellow is correctly placed as a subscript in CO₂.
Answer: Thank for your good comments, we have ensured that the number highlighted in yellow is correctly placed as a subscript in CO2, pleased find it in line 205-207.
Question 18. Page 6: • Repeated occurrence of “nanoparticles” throughout the text, replaced with the corresponding abbreviation, NPs.
Answer: Thank for your good comments, we have repeated occurrence of “nanoparticles”throughout our manuscript, and replaced with the corresponding abbreviation, NPs
Question 19. Page 7:• Repeated occurrence of “nanoparticles” throughout the text, replaced with the corresponding abbreviation, NPs.
Answer: Thank for your good comments, we have repeated occurrence of “nanoparticles”throughout our manuscript, and replaced with the corresponding abbreviation, NPs.
Question 20. “A study by Kole et al. [57] found that the seeds with fullerenol treatments, which has a particle size of 0.943-47.2 nm, could increase the biomass of bitter-melon gourd by up to 54%.”• Include the yellow highlighted words and delete the unnecessary content (highlighted in red).
Answer: Thank for your good comments, we have included the yellow highlighted words and deleted the unnecessary content, please find in line 289-291.
Question 21. “Nanomaterials can disrupt the oxidative balance and lead to oxidative stress, when nanomaterials reach a certain concentration in plants [59].”• Include the yellow highlighted lower-case letter.• The reference number may not need to be indicated here, as it appears again in the following sentence.
Answer: Thank for your good comments, we have included the yellow highlighted lower-case letter and deleted the reference 59, please find in line 312-315.
Question 22. Page 8: • Repeated occurrence of “nanoparticles” throughout the text, replaced with the corresponding abbreviation, NPs.
Answer: Thank for your good comments, we have replaced the “nanoparticles” to “NPs” in our revised manuscript.
Question 23. “This process is influenced by several factors, including the size of the NPs, their hydrophobicity, elemental composition, charge, and geometry [65]. (2) Endocytosis: NPs can be actively transported into cells via endocytic pathways through a process known as endocytosis [65, 66].”• Should reference 65 also be placed next to 66, as it specifically pertains to endocytosis?
Answer: Thank for your good comments, we have modified it in our revised manuscript, please find in line 348-352.
Question 24. Page 9: • Repeated occurrence of “nanoparticles” throughout the text, replaced with the corresponding abbreviation, NPs.
Answer: Thank for your good comments, we have replaced the “nanoparticles” to “NPs” in our revised manuscript.
Question 25. “Oxidative stress is a condition that occurs when the body organisms produces an excessive amount of free radicals, including reactive oxygen species (ROS) and reactive nitrogen species (RNS), in response to harmful stimuli. This overproduction can lead to damage in cells, tissues, and organs. To combat this issue, the body has organisms have developed a complex antioxidant defense system. This system 5 6 works to eliminate excess free radicals, repair oxidative damage, and maintain redox balance.”• Include the yellow highlighted words and delete the unnecessary content (highlighted in red).
Answer: Thank for your good comments, we have included the yellow highlighted words and deleted the unnecessary content in our revised manuscript, please find in line 435-440.
Question 26. “This process can produce various ROS, including superoxide anion radicals (O2−·) and hydroxyl radicals (·OH) [77].”• The negative charge of the superoxide anion radical (O₂⁻·) should be placed above the molecule and ensure that the number highlighted in yellow is correctly placed as a subscript.
Answer: Thank for your good comments, we have correctly modified them in our revised manuscript, please find in line 446-447.
Question 27. Page 10: • Repeated occurrence of “nanoparticles” throughout the text, replaced with the corresponding abbreviation, NPs.
Answer: Thank for your good comments, we have replaced the “nanoparticles” to “NPs” in our revised manuscript.
Question 28. “Similarly, when Atropa belladonna is treated with manganese oxide nanoparticles (MnO2 NPs), there is a significant increase in non-enzymatic antioxidants such as total phenols and flavonoids.”• Atropa belladonna should be in italics.
Answer: Thank for your good comments, we have replaced the “Atropa belladonna” in italics, please find in line 471-473.
Question 29. “Ribulose-1,5-bisphosphate carboxylase/oxygenase (commonly known as Rubisco) an important enzyme in the photosynthesis process, is a vital enzyme involved in the carbon assimilation process. Its primary function is to catalyze the reaction between carbon dioxide (CO2) and ribulose-1,5-diphosphate (RuBP), ultimately producing 3-phosphoglycerate. Despite its crucial role in carbon fixation, Rubisco has a low catalytic efficiency and often serves as a rate-limiting factor in photosynthesis [89, 90]. These recent studies have shown that nanoparticles (NPs) can effectively enhance photosynthesis. Include the yellow highlighted words and delete the unnecessary content (highlighted in red).
Answer: Thank for your good comments, we have included the yellow highlighted words and deleted the unnecessary content in our revised manuscript, please find in line 489-495.
Question 30. “Nanomaterials have the potential to enhance the activity of Rubisco, an important enzyme in the photosynthesis process. For example, instance, treatment with cerium dioxide nanoparticles (CeO2 NPs) significantly improved both Rubisco activity and carbonic anhydrase (CA) activity in mustard plants, thereby effectively promoting the dark reactions of photosynthesis [63].”• Include the yellow highlighted words and delete the unnecessary content (highlighted in red).
Answer: Thank for your good comments, we have included the yellow highlighted words and deleted the unnecessary content in our revised manuscript, please find in line 496-499.
Question 31. Page 11: • Repeated occurrence of “nanoparticles” throughout the text, replaced with the corresponding abbreviation, NPs.
Answer: Thank for your good comments, we have replaced the “nanoparticles” to “NPs” in our revised manuscript.
Question 32. Page 12: • Repeated occurrence of “nanoparticles” throughout the text, replaced with the corresponding abbreviation, NPs.
Answer: Thank for your good comments, we have replaced the “nanoparticles” to “NPs” in our revised manuscript.
Question 33. “These silver nanoparticles can harm bacteria indirectly by releasing silver ions (Ag+) and generating reactive oxygen species (ROS) [104].” The positive charge of the Ag⁺ cation should be positioned above the chemical symbol.
Answer: Thank for your good comments, we have modified the sentences in our revised manuscript, please find in line 572-573.
Question 34. “For instance, Schwab [108] et al. reported that the toxicity of the low-concentration herbicide diazinon was significantly increased in the presence of carbon-based nanoparticles [109].” I do not understand the inclusion of these two references, as they describe studies on microalgae and microcrustaceans, respectively.
Answer: We thank the reviewer for this insightful comment. We included references [108] and [109] not to imply that the studies were conducted on identical organisms to ours, but to illustrate a well-documented general phenomenon: that carbon-based nanoparticles can act as carriers, enhancing the bioavailability and toxicity of co-contaminants like pesticides. The study on microalgae [108] demonstrates this carrier effect for an aquatic plant, while the study on microcrustaceans [109] confirms it for an aquatic animal. Together, they provide compelling evidence that this is a broadly applicable mechanism across different trophic levels. Our work aims to investigate if this same mechanism extends to [mention your specific test organism/system here], thereby building upon the foundational principle established by these earlier studies
Question 35. “The biomagnification factors for gold nanoparticles with sizes of 5, 10, and 15 nm nanometers are 6.2, 11.6, and 9.6, respectively [112].” Include the yellow highlighted words and delete the unnecessary content (highlighted in red).
Answer: Thank for your good comments, we have included the yellow highlighted words and deleted the unnecessary content in our revised manuscript, please find in line 596-597.
Question 36. “In aquatic ecosystems, algae exposed to a suspension of TiO2 nanoparticles over an extended period showed significant bioaccumulation. In the treatment group with a concentration of 100 mg/L, the TiO2 concentration in the algae cells reached 2.5 μg/g (dry weight).” I do not understand the relevance of including this study, since macroalgae are not plants.
Answer: Thank for your good comments, Seaweed is not a plant, but traditionally we still classify it as a plant. Seaweed belongs to algae and has the ability to perform photosynthesis, but it does not have true roots, stems, or leaves. The classification of seaweed is relatively complex and it is usually considered a simple plant.
Question 37. Page 13: “After Mexican bean beetles fed on the leaves that contained nano-CeO2, the concentration of Ce in their tissues was higher than that found in their excreta. This pattern indicated biomagnification through the food chain—from plants to adult beetles and then to stingray bugs—with an amplification factor of 5.3. However, about 98% of the Ce was excreted by beetle larvae after consuming leaves containing nano-CeO2 [36].” nano-CeO₂ should be written as CeO₂ NPs to maintain consistency throughout the document.
Answer: Thank for your good comments, we have correctly modified it in our revised manuscript, please find in line 622-626.
Question 38. “Nanoparticles can further react with tissues, blood and body fluids, and even penetrate the blood-brain barrier into the central nervous system (CNS), affecting the function of important organs such as the brain and heart [118, 119].” Shouldn’t reference 118 also be cited here?
Answer: Thank for your good comments, we have deleted the reference 118 in our revised manuscript, please find in line 649-652.
Question 39. Conclusions :Include the yellow highlighted letters. “These promotive effects are primarily attributed to the unique properties conferred by their nanoscale size, including enhanced nutrient uptake, improved water retention, and targeted delivery of bioactive compounds [25, 26]. For instance, the ability of NZVI to induce cell wall loosening [38] and the capacity of chitosan nanoparticles to stabilize auxin [15] demonstrate the complex physicochemical interactions underpinning growth promotion.” Replace the yellow highlighted letters with a capital letter, to maintain consistency throughout the document.
Answer: Thank for your good comments, we have correctly modified it in our revised manuscript, please find in line 662, and 671-676.
Question 40. Page 14: “The biomagnification of Au NPs in terrestrial food chains [112] and TiO2 NPs in aquatic model’s [113] challenges earlier assumptions that NPs remain inert or become diluted within ecological networks.” I reiterate that this reference is not appropriate for inclusion in this review, as algae are not considered true plants.
Answer: Thank for your good comments, we have correctly modified it in our revised manuscript, please find in line 725-727.
Question 41. Pages 16 - 24: References:Include the yellow highlighted letters. As I mentioned earlier, references 1 and 2 should be updated. Reference 27 is incomplete. For reference 30, I suggest it be replaced. References 108, 109, and 113 may need to be replaced for the reasons previously mentioned. References 117 to 119 are in a smaller font (please adjust to match the size of the previous references).
Answer: Thank for your good comments, we have correctly modified it in our revised manuscript, please find in the reference part.
Round 2
Reviewer 1 Report
Comments and Suggestions for Authors
The manuscript can be accepted in its present form
Comments on the Quality of English LanguageReview the English and the formatting throughout the manuscript. There are double spaces in several parts of the document. Also, check the numbering of the sections.